# Linking Categorical and Dimensional Approaches to Assess Food-Related Emotions

**DOI:** 10.3390/foods11070972

**Published:** 2022-03-27

**Authors:** Alexander Toet, Erik Van der Burg, Tim J. Van den Broek, Daisuke Kaneko, Anne-Marie Brouwer, Jan B. F. Van Erp

**Affiliations:** 1TNO Human Factors, Netherlands Organization for Applied Scientific Research, Kampweg 55, 3769 Soesterberg, The Netherlands; erik.vanderburg@tno.nl (E.V.d.B.); d.kaneko@kikkoman.nl (D.K.); anne-marie.brouwer@tno.nl (A.-M.B.); jan.vanerp@tno.nl (J.B.F.V.E.); 2Brain and Cognition Department, University of Amsterdam, 1012 Amsterdam, The Netherlands; 3TNO, Netherlands Organization for Applied Scientific Research, Research Group Microbiology & Systems Biology, 3700 Zeist, The Netherlands; tim.vandenbroek@tno.nl; 4Kikkoman Europe R&D Laboratory B.V., Nieuwe Kanaal 7G, 6709 Wageningen, The Netherlands; 5Research Group Human Media Interaction, University of Twente, 7522 Enschede, The Netherlands

**Keywords:** emotions, emotion terms, food images, EmojiGrid, EsSense25, valence, arousal

## Abstract

Reflecting the two main prevailing and opposing views on the nature of emotions, emotional responses to food and beverages are typically measured using either (a) a categorical (lexicon-based) approach where users select or rate the terms that best express their food-related feelings or (b) a dimensional approach where they rate perceived food items along the dimensions of valence and arousal. Relating these two approaches is problematic since a response in terms of valence and arousal is not easily expressed in terms of emotions (like happy or disgusted). In this study, we linked the dimensional approach to a categorical approach by establishing mapping between a set of 25 emotion terms (EsSense25) and the valence–arousal space (via the EmojiGrid graphical response tool), using a set of 20 food images. In two ‘matching’ tasks, the participants first imagined how the food shown in a given image would make them feel and then reported either the emotional terms or the combination of valence and arousal that best described their feelings. In two labeling tasks, the participants first imagined experiencing a given emotion term and then they selected either the foods (images) that appeared capable to elicit that feeling or reported the combination of valence and arousal that best reflected that feeling. By combining (1) the mapping between the emotion terms and the food images with (2) the mapping of the food images to the valence–arousal space, we established (3) an indirect (via the images) mapping of the emotion terms to the valence–arousal space. The results show that the mapping between terms and images was reliable and that the linkages have straightforward and meaningful interpretations. The valence and arousal values that were assigned to the emotion terms through indirect mapping to the valence–arousal space were typically less extreme than those that were assigned through direct mapping.

## 1. Introduction

### 1.1. Categorical and Dimensional Food-Elicited Emotion Assessment

One of the major challenges in food marketing and consumer studies is to understand the motivations that drive consumer choices. It appears that emotions play an essential role in the food choices of consumers [1,2]. As a result, knowledge of hedonic (liking) ratings alone is not sufficient to accurately predict food choice behavior [3,4].

Despite its long history, emotion research still has not reached a consensus on the fundamental nature of emotions [5,6,7]. The two main prevailing and opposing views are (1) the discrete or categorical approach, which assumes that there are a limited number of discrete, universal emotions (possibly subserved by different brain mechanisms) [8,9] and (2) the constructionist approach, which is based on the assumption that the brain constructs emotions along a few continuous dimensions (e.g., valence and arousal) [10]. These different conceptualizations of emotions have resulted in different methodologies that may be used to investigate emotion perception.

A widely used categorical approach to assess emotional responses to food and beverages is through lexicon-based (verbal) tools that enable users to select and/or rate the words that best express their food-related feelings [11]. A popular and standardized lexicon of food-related affective terms that is used for this purpose is the EsSense Profile (39 terms: [12]). Users report their food-related emotional response with the EsSense Profile by rating each of its 39 terms on a 5-point intensity scale [1,2,13]. Next to emotions, the EsSense Profile also includes diffuse affective states such as moods, which are characterized by prolonged subjective feelings (e.g., “loving” or “affectionate”; [14]). When used in a check-all-that-apply (or CATA) paradigm, users are instructed to select those EsSense Profile terms that apply to the focal sample [15]. Alternatively, in a rate-all-that-apply (or RATA) paradigm, users are instructed to also rate the extent to which each selected term applies to the focal sample [16].

Although widely used, it has been argued that lexicon-based tools force people to express their feelings through a limited set of prescribed words, resulting in rationalized answers that do not necessarily reflect the unconscious influences that play a major role in emotional food perception [17]. Moreover, people typically find it difficult to express or verbalize their emotions (especially for mixed or complex ones) and the labels (emotion terms) that are provided to describe them are often inherently ambiguous [17,18] or even perceived as strange or irrelevant in a food-related context [19]. These considerations have stimulated the development of graphical (non-verbal) tools that allow users to report their feelings in a more intuitive manner by indicating or rating the figures that best represent their current affective state (for a review see [20]). It has, for instance, been shown that people reliably and intuitively use emoji (facial icons) to report their food-related emotional experiences [21,22,23,24,25,26].

Next to the lexicon-based (categorical) approach, there is also a dimensional approach to the assessment of (food-related) emotions. In this approach, valence (the pleasantness; the degree of positive or negative affective response to a stimulus) and arousal (the intensity of the affective response to a stimulus) are adopted as the principal dimensions of emotions in general (the circumplex model of human core affect [8,27]) and of food-evoked emotions in particular [13]. Valence and arousal both play a distinct and critical role in eating-related behavior [28]. A parsimonious graphical (language independent) self-report tool that was specifically developed for this dimensional approach to the affective appraisal of food is the EmojiGrid: a square grid (resembling the Affect Grid [29]) that is labeled with emoji expressing different degrees of valence and arousal ([20]; see Figure 1; see also https://en.wikipedia.org/wiki/EmojiGrid (accessed on 4 March 2022). The EmojiGrid enables its users to intuitively report their affective state with a single response by placing a checkmark at the appropriate location on the grid [11,20]. It has been observed that verbal labeling diminishes one’s response to affective stimuli [30]. Intuitive (graphical) self-report tools that limit analytical thinking may therefore be preferred for measuring (food-related) emotions since they may tap more directly into the irrational and non-cognitive thought processes that are involved in food choice than verbal methods can [17]. Hence, tools like the EmojiGrid may yield responses that more closely reflect the truly experienced emotions than self-reports that are obtained with verbal tools [2].

The dimensional and categorical approaches are complementary. While the parsimonious dimensional approach using the EmojiGrid affords an intuitive and efficient method to assess food-related emotions in terms of their resultant valence and arousal, it does not provide a description of the response in terms of discrete or basic emotions [31], unlike a lexicon-based categorial approach. Although most people will understand what it means that some kind of food is rated as either positive or negative (valence) and that it evokes a given amount of arousal, the corresponding point in the two-dimensional valence–arousal space has no specific meaning and cannot easily be communicated [32]. For instance, it would seem weird to tell someone that you feel 2.7 positive and 1.4 aroused. Combining the two approaches could allow one to quickly zoom in on the most salient features of a response (valence and arousal), followed by a further, more detailed inspection of the distinct underlying factors (emotions) that are contributing to that response and to express these in words [33].

### 1.2. Related Work

Using a CATA paradigm with cashew nuts, peanuts, chocolate, fruit and processed tomatoes as the focal product categories, participants in a study by Jaeger, Spinelli, Ares and Monteleone [34] reported their sensory product perceptions (in terms of sensory attributes like appearance, flavor, taste, texture and odor) and associations with emotion terms (from the EsSense Profile). Relationships between the resulting food-elicited emotional associations and the sensory terms were established by mapping both to the circumplex model of human core affect through correspondence analysis. While many of these relationships were easy to interpret, others were less obvious. Jaeger et al. [34] suggested to further validate their mapping of emotion terms to the valence–arousal space through a direct mapping procedure.

Scherer et al. [32] linked a dimensional and a categorical approach to emotion assessment through the Geneva Emotion Wheel (GEW) graphical response tool. In the GEW, 20 emotion terms are equidistantly spaced around the circumference of a circular two-dimensional space representing the dimensions of valence and control/power. Different emotion terms (that only appear when the user moves a cursor over their position) are placed inside the circle, such that their intensity increases with their distance from the center. Thus, the GEW combines three response dimensions (i.e., valence, intensity, and control/power) in a two-dimensional representation. However, the control/power dimension is rather abstract and appears difficult for users to rate. Although the GEW was developed as a general instrument for the measurement of emotional response to affective stimuli, its emotion terms do not apply to food-elicited emotions. Also, arousal is not explicitly measured. Although arousal and intensity are related, both are distinct concepts, which are not linearly related [35]

Lorette [6] linked the discrete categorical approach to the continuous dimensional approach through a two-step instrument called the Two-Dimensional Affect and Feeling Space (2DAFS). The 2DAFS is a clickable and labeled affect grid that is followed by the presentation of a valence–arousal space that is labeled with 36 basic emotion terms. The emotion terms were centered at valence and arousal coordinates that had been determined in previous (unrelated) studies in which these terms had been rated for their valence and arousal [36,37,38]. After reporting their appraisal (in terms of valence and arousal) of the emotional stimuli by clicking on the affect grid, the users can further categorize their response by selecting one or more words from the spatially ordered set of emotion terms. Since the emotion terms are positioned according to their valence and arousal ratings, terms that probably apply most (and are therefore most likely to be selected by the user) are arranged closest to the location where the user clicked on the grid, enabling an efficient and fast response. Although the 2DAFS has been developed as a general instrument to measure emotional responses to affective stimuli, its emotion terms are not suitable for use to characterize food-elicited emotions. A further limitation of the instrument is that participants can only choose one emotion term per response, thus preventing the reporting of mixed emotions.

### 1.3. Linking the Different Approaches

The goal of the present study was to combine the parsimony of a single-response dimensional (EmojiGrid) approach with the specificity of lexicon-based tools by linking food-related emotion terms to the two-dimensional valence–arousal space. Through such relations, EmojiGrid responses can be expressed in terms of (weighted combinations of) different discrete emotions. The availability of a language-independent dimensional single-response tool like the EmojiGrid that also affords a categorical verbal output will be beneficial for cross-cultural studies and for studies involving children or low-literate people.

## 2. Methods and Procedures

### 2.1. Overview of the Approach

The participants in this study performed four different tasks (see Figure 2). In order to investigate the mapping between food images and the valence–arousal space, the participants rated their emotional responses to different food images using the EmojiGrid (*Image2Grid* task: matching each food image to the most appropriate location on the EmojiGrid [11,20]). To establish the mapping between the emotion terms and the food images, the participants (a) matched each food image to the most appropriate labels from a set of simultaneously presented emotion terms (*Image2Label* task: matching food images to emotion terms) and (b) assigned each of these emotion terms to a selection from a set of simultaneously presented food images (*Label2Image* task: labeling food images with emotion terms). To establish a direct mapping between the emotion terms and the valence–arousal space, the participants attributed each emotion term to the most appropriate location on the EmojiGrid (*Label2Grid* task: labeling the EmojiGrid with emotions terms).

Note that the mappings between the emotion terms and the food images that result from the matching and labeling tasks need not be identical. Also, the EmojiGrid positions (the valence and arousal values) that are assigned to the terms and images through the matching and labeling tasks need not be the same. In the ‘matching’ conditions, the participants first imagined how the food that was shown in the image would make them feel and then selected either the emotion terms or the EmojiGrid position that best described their feelings that are associated with that food. In the ‘labeling’ conditions, the participants first imagined how a given emotion term felt and then they selected either the foods (images) that appeared capable to elicit that feeling or reported the combination of valence and arousal (using the EmojiGrid) that best reflected that feeling. As a result, the mapping between the emotion terms and images need not be bidirectionally identical. The process of matching yielded the assignment of each food image to those emotion terms that are most characteristic for that image, while the process of labeling yielded the assignment of each label to all of the food images to which it may apply to some degree. Hence, the labels that are assigned to an image need not correspond to the emotions that are intuitively and most intensely experienced (that first come to mind) when seeing that image. Testing both the matching (*Image2Label* task) and mapping (*Label2Image* task) conditions in this study served to assess the reliability (association strength) of the relation between the emotion terms and the food images.

In addition to the direct mapping between the emotion terms and the valence–arousal space (EmojiGrid), as established through the *Label2Grid* task, we also established an indirect mapping by combining the results of the *Image2Grid* task with those of the *Image2Label* task. This was done by assigning the mean valence and arousal ratings (mean EmojiGrid coordinates) that were reported for a given image in the *Image2Grid* task to the emotion terms that were assigned to that image in the *Image2Label* task. In the rest of this paper, we will refer to this indirect mapping of the emotion terms to the EmojiGrid as the *Label2Image2Grid* mapping.

### 2.2. Participants

A total of 480 English-speaking participants from the UK (240 female, mean age = 26.1 years, SD = 5.1, range = 18–40 years) were recruited via the Prolific database (https://prolific.ac, accessed on 4 March 2022). The exclusion criterion was (color) vision deficiency.

The experimental protocol was reviewed and approved by the TNO Internal Review Board (approval code: 2019-033, approval date: 10 May 2019). The study was conducted in accordance with the Helsinki Declaration of 1975, as revised in 2013 [39]. Participation in this study was voluntary. The participants received financial compensation for their participation.

### 2.3. Stimuli

#### 2.3.1. Food Images

Twenty food images (Figure 3) were selected from the Cross-Cultural Food Image Database (CROCUFID [40]). CROCUFID includes high-resolution images of sweet, savory, natural, and processed food from Western and Asian cuisines, photographed according to a standard protocol, so that all of the food items were observed against the same background (a white plate) and from a fixed viewing angle (45°). The 20 images that were used in this study were selected such that their associated mean valence ratings covered the entire valence scale [11,20]. They represent natural food (e.g., fruit and salad), processed food (e.g., cakes and a burger) and rotten or molded food (e.g., rotten pears and molded salad). In the check-all-that-apply (CATA) labeling procedure that was used in this study (see Section 2.6.3) all of the 20 food images were simultaneously presented in a 5 × 4 rectangular grid layout. For half of the participants, the order was left–right and for the remaining participants the order was up–down (i.e., reversed). This was done in order to neutralize any selection biases that could arise when the participants scanned the image matrix in reading order (left to right and top to bottom), paying more attention to the terms at the top of the list than to those at the bottom. The 20 food images are provided with detailed information in the Appendix A.

#### 2.3.2. Emotion Terms

Twenty-five emotion terms were used in this study, all from the EsSense25 lexicon [41] (a reduced version of the EsSense Profile [12]). The EsSense25 is a validated list of food-specific emotion terms that is used to measure self-reported food-evoked emotional associations [15]. In the check-all-that-apply (CATA) labeling procedure that was used in this study (see Section 2.6.3 and Section 2.6.4), the 25 emotion terms of the EsSense25 were presented in alphabetical order to the first half of the participants and in reversed alphabetical order to the second half (see Figure 4). This was done to neutralize any selection bias that could arise when participants scanned the list in reading order, paying more attention to the terms that were at the top of the list than those that were at the bottom [42].

### 2.4. Measures

#### 2.4.1. Demographics

The participants in this study reported their age and gender.

#### 2.4.2. Valence and Arousal

In accordance with the circumplex model of affect [8], the affective responses that are elicited by food-related stimuli vary mainly over the two principal affective dimensions of valence (i.e., pleasure or displeasure) and arousal (i.e., activation or deactivation). In this study, valence and arousal were measured with the EmojiGrid graphical self-report tool [20]. The EmojiGrid is a square grid that is labeled with emoji that express various degrees of valence and arousal (Figure 1). Users rate their affective appraisal of a given stimulus by pointing and clicking at the location on the grid that best represents their impression in terms of valence and arousal. The EmojiGrid was inspired by Russell’s Affect Grid [29] and was originally developed and validated for the affective appraisal of food stimuli [11,20], since conventional affective self-report tools (e.g., Self-Assessment Manikin [43]) are frequently misunderstood in that context [11,20]. It has since also successfully been used and validated for the affective appraisal of a wide range of different emotional stimuli, such as images [44], sound and video clips [45], touch events [46], odors [47,48,49] and VR experiences [50]. Since it is intuitive and language-independent, the EmojiGrid is also suitable for cross-cultural research [11,51] and research involving children or low-literate participants.

### 2.5. Data Analysis

The statistical data analyses were conducted using IBM SPSS Statistics 26 for Windows (IBM, New York, USA), R software version 4.1.1 (The R Foundation for Statistical Computing), and the Python programming language version 3.9 (The Python Software Foundation). Descriptive statistics were used in order to calculate (1) the percentage of the emotion terms that were selected for each food image and (2) the percentage of the food images that were selected for each emotion term. The intraclass correlation coefficient (ICC) estimates and their 95% confident intervals were based on a mean-rating (k = 3), absolute agreement, 2-way mixed-effects model [52,53]. ICC values less than 0.5 are indicative of poor reliability, values between 0.5 and 0.75 indicate moderate reliability, values between 0.75 and 0.9 indicate good reliability and values greater than 0.9 indicate excellent reliability [52]. For all of the other analyses, a probability level of *p* < 0.05 was considered to indicate statistical significance.

### 2.6. Procedure

Participants took part in an anonymous online survey that was created with the Gorilla Experiment Builder [54]. The survey commenced by presenting general information about the experiment and thanking the participants for their contribution. The participants were informed that during the experiment they would be asked to report their first impressions of 20 food images (e.g., by imagining how consuming the food that was shown in the images would make them feel) and 25 food-related words. It was emphasized that there were no correct or incorrect answers and that it was important to respond seriously. Subsequently, the participants signed a digital informed consent, affirming that they were at least 18 years old and voluntarily participating in the study. The survey then continued with an assessment of the demographics (age and gender) of the participants. The main body of the survey consisted of four tasks that were performed in a fixed order (see Figure 5). Two of these tasks were labeling tasks (the blue arrows in Figure 2) in which the participants assigned the EsSense25 terms to the food images and to the EmojiGrid. The other two tasks were matching tasks (the red arrows in Figure 2) in which the participants matched the food images to the EsSense25 terms and to the EmojiGrid. These four experimental tasks are described in further detail in the next four subsections. The participants received visual feedback about their progress through the experiment via a blue progress bar in the lower part of the screen. They could take a short break between the tasks. To assess the seriousness of the participation, we included a validated seriousness check at the end of the experiment (asking the participants if they had answered seriously, per [55]). The average duration of the experiment was about 15 min.

#### 2.6.1. Task I: *Image2Grid*

In the first task, *Image2Grid*, each trial showed a randomly selected food image (from the total set of 20 stimuli) next to the EmojiGrid (Figure 6). The participants were asked to report how each image made them feel by using the EmojiGrid. Clicking on the EmojiGrid initiated the next trial. The participants first read a brief explanation about the use of the EmojiGrid response tool. Then they performed two practice trials (using two food images that were not included in the stimulus set) in order to familiarize themselves with the use of this tool. Immediately after these practice trials, the actual rating experiment started. For each of the 20 trials, a different food image was presented and the participants reported their affective appraisal of the food that was shown by clicking on the EmojiGrid. The task was self-paced.

#### 2.6.2. Task II: *Image2Label*

In the second task, *Image2Label*, each trial showed a randomly selected food image next to all of the EsSense25 terms (Figure 7). For each food image, the participants were asked to click on all of the terms that best described how the image made them feel (a CATA procedure). When selected, the EsSense25 terms became highlighted. After selecting all of the terms that they considered to apply to the image that was shown, the participants could start the next trial by clicking on a “next” button. The participants performed two practice trials (using two food images that were not included in the stimulus set) in order to familiarize themselves with the EsSense25 terms and the procedure. Immediately after these practice trials, they performed 20 experimental trials. On each of these trials, a different food image was presented and the participants clicked on the emotion terms that in their opinion best described how that image made them feel. The task was self-paced. 

#### 2.6.3. Task III: *Label2Image*

In the third task, *Label2Image*, each trial showed a randomly selected EsSense25 term next to all 20 of the food images (Figure 8). On each of the 25 trials, the participants were asked to click on all of the food images to which the current emotion term applied (i.e., a CATA procedure). The food images that were selected became highlighted. After selecting all of the relevant images, the participant could start the next trial by clicking on a “next” button. The task was self-paced. Note that the *Label2Image* task yields a mapping between images and emotion terms that is the inverse of the mapping that results from the *Image2Label* task. As mentioned in the Introduction, the rationale for including this task is to investigate the reliability (the association strength) of the mapping between the images and emotion terms.

#### 2.6.4. Task IV: *Label2Grid*

In the fourth task, *Label2Grid*, each trial showed a randomly selected EsSense25 term next to the EmojiGrid (Figure 9). On each of the 25 trials, the participants were asked where they would click on the EmojiGrid in order to respond that a given food would make them feel like the term shown. Clicking on the EmojiGrid started the next trial. The task was self-paced.

## 3. Results

In response to the seriousness check, all of the participants reported that they had answered all of the questions seriously. No participants were excluded from the analysis.

### 3.1. Task I: Image2Grid

In order to evaluate the face validity of the valence and arousal ratings that were collected for the food images, we ordered the food images based on their mean valence ratings (from low to high valence). As expected, Table 1 shows that the highest mean valence ratings were obtained for the images of fresh fruit (the apple, orange and strawberries) and pastries, while neutral ratings were obtained for images of boiled eggs and salads and the lowest mean valence ratings correspond to images of molded food (the molded salads, banana and pear).

To quantify the agreement between the mean valence and arousal ratings that were obtained in the present study and those that were reported previously by Kaneko et al. [11], we computed the intraclass correlation coefficients (ICC) with their 95% confidence intervals for the mean valence and arousal ratings that were obtained in both studies. The ICC value for valence is 0.995 [0.988–0.998] while the ICC for arousal is 0.980 [0.949–0.992], indicating that the mean valence and arousal values that were measured in both studies are in excellent agreement.

### 3.2. Task II: Image2Label

Figure 10 (filled diamonds) shows the percentage of the participants that linked each image to each of the different EsSense25 terms in the *Image2Label* task. The emotion terms that are predominantly related to pleasure (good, pleasant and happy) and displeasure (worried and disgusted) were the most frequently used. The emotion terms expressing different degrees of (de-)activation (arousal) were also used throughout this study (e.g., calm, bored, interested and enthusiastic), indicating their relevance for characterizing food-related experiences. As expected, the food images that were overall rated low on valence were most frequently labeled with negative terms (e.g., aggressive, worried and disgusted), while the images that were overall rated high on valence were most frequently labeled with positive terms (e.g., happy, pleasant and good). The items that were rated near-neutral on valence (the boiled eggs, salads and cucumber) were frequently labeled with neutral terms (e.g., bored or mild).

### 3.3. Task III: Label2Image

Figure 10 (open diamonds) shows the percentage of the participants that labeled each food image with each of the EsSense25 terms in the *Label2Image* task. The results are highly similar to those of the *Image2Label* task: the images that were overall rated low on valence were most frequently labeled with negative terms (e.g., aggressive, worried and disgusted), while the images that were overall rated high on valence were most frequently labeled with positive terms (e.g., happy, pleasant and good). Overall, the emotion terms were used more frequently to label the images that were presented in the *Label2Image* task than those which were presented in the *Image2Label* task (i.e., the open symbols typically have a larger area than the filled symbols).

In order to investigate the reliability (association strength) of the associations between the emotion terms and food images, we computed the Pearson correlation between the image-to-term assignment frequencies that were obtained from the *Image2Label* task and the term-to-image assignment frequencies that were obtained from *Label2Grid* task. The Pearson correlation coefficient was 0.84 (with a 95% CI of [0.81, 0.86]), indicating that the mapping was highly reliable.

### 3.4. Task IV: Label2Grid

Figure 11A illustrates the distribution of the responses that were made by the participants when they were matching the emotion terms directly to the EmojiGrid in the *Label2Grid* task. Here the data is only shown for the terms disgusted, happy, guilty and understanding. The distributions for the remaining emotion terms are provided in the Appendix A.

By combining the results from the *Label2Image* and *Image2Grid* tasks, it was also possible to establish an indirect mapping of the emotion terms to the valence–arousal space. This was done by computing, for each emotion term, the average valence and arousal values over all of the images (as were determined in the *Image2Grid* task) to which this term had been assigned (in *Label2Image* task). Figure 11B shows that, for most of the emotion terms, indirectly mapping to the valence–arousal space via the food images yielded a spatial distribution of the responses that is similar to the distribution that resulted from the direct mapping. The white crosses in Figure 11 represent the group mean arousal and valence ratings. Each row in Figure 11 represents the results for the four different words (disgusted, happy, guilty and understanding).

Two tailed *t*-tests were conducted in order to examine whether the mean valence and mean arousal ratings for each emotion term were significantly different between the indirect (*Label2Image2Grid*) and direct (*Label2Grid*) mappings. Table 2 illustrates an overview of these analyses. The significant effects are shown in bold font for illustrative purposes.

The Wilcoxon signed-rank tests yielded a significant difference between the indirect (*Label2Image2Grid*) and the direct (*Label2Grid*) mappings for most of the terms (Table 2). In fact, for all of the terms that were used we found a significant difference between either the mean valence or the mean arousal ratings. Hence, the valence and arousal ratings that the participants assigned to the emotion terms were not consistent with their valence and arousal ratings for the food images to which they assigned these terms. In general, the emotion terms were rated more extremely on valence and arousal than the food images. This is also clear from Figure 11B (the direct mapping) where the response distributions are closer to the edges of the valence–arousal space than the ratings that were indirectly obtained via the food images (see the Appendix A).

Figure 12 shows the average locations of the emotion terms in the valence–arousal space, determined both through direct (the red squares) and indirect (the blue circles) mapping. This figure shows that the directly mapped terms are overall located further towards the periphery of the valence–arousal space, while the indirectly mapped terms are located more centrally. Thus, it appears that the affective strength (experienced intensity) of all of the terms is more extreme when it is measured by direct mapping than when it is measured by indirect mapping. Interestingly, the average location of the term guilty resulting from the indirect (*Label2Image2Grid*) mapping is the opposite of its position resulting from the direct (*Label2Grid*) mapping.

Figure 12 also shows the superposition of the circumplex model of human core affect by Yik et al. [27] over the two-dimensional valence–arousal space. This model defines 12 domains (numbered from 1 to 12, following [34]) that represent (a) the poles of the two core dimensions (“pleasure–displeasure” and “activation–deactivation”) and (b) eight emotional domains that are defined as a combination of both of the core dimensions. For convenience, these domains are taken to be of equal angular extent (30°), although this is not required for a circumplex model [27]. Using a questionnaire-based approach, Jaeger et al. [34] established the linkages between food-elicited emotional associations (the EsSense Profile) and sensory characteristics by mapping both to the 12 domains of the circumplex model through correspondence analysis. To compare our present results with those of Jaeger et al. [34] we computed, for each emotion term, its domain-membership function as one plus the radial angle of the term’s location on the EmojiGrid divided by the width of a domain (30°), where the radial angle increases clockwise, starting at 0 for the activation dimension. This definition of the membership function results in fractional values, with fractions smaller (or larger) than 0.5, indicating that the emotion term has aspects in common with the previous (next) domain. Note that the membership function that was used by Jaeger et al. [34] used only multiples of 0.5. Table 3 lists the mapping between the EsSense25 emotion terms and the 12 domains of the core affect model that was presented by Yik et al. [27], from the study by Jaeger et al. [34] and obtained in this study through both direct mapping (*Label2Grid*) and indirect mapping (*Image2Label2Grid*). The Pearson correlation between the direct and indirect mapping of emotion terms to the domains of the circumplex model is 0.96, indicating that the nature of the emotional appraisals remains constant between the indirect and direct mappings while only their intensity varies (with ratings for the direct mapping being more extreme). The Pearson correlation between the indirect mapping that was obtained in this study and the mapping that was reported by Jaeger et al. [34] is 0.81, indicating a strong agreement between both results. The largest differences between both studies are found for the terms active and guilty. While active was mapped to the ‘pleasant activation’ domain by Jaeger et al. [34] it was mapped to the ‘pleasure’ domain in the current study. Guilty was mapped to the ‘activated displeasure’ domain in the study by Jaeger et al. [34] and to the ‘activated pleasure’ domain in this study, being even the opposite on the valence dimension.

As observed by Jaeger et al. [34] and acknowledged by Meiselman [56], Figure 12 also shows that the EsSense25 is quite unbalanced and lacks emotion terms with a negative valence (e.g., a domain like ‘unpleasant activation’ is not represented).

## 4. Discussion

In this study we established a link between a categorial (lexicon-based) tool (the EsSense25) and a dimensional (valence and arousal-based) tool (the EmojiGrid) in order to assess food-related emotions. To establish a mapping between the 25 emotion terms of the EsSense25 and the set of 20 food images, the participants labeled each food image with a subset of the emotion terms (*Label2Image* task) and mapped both the food images and the emotion terms to the valence–arousal space (the *Image2Grid* and *Label2Grid* tasks, both using the EmojiGrid). By combining (1) the mapping between the emotion terms and the food images with (2) the mapping of the food images to the valence–arousal space, we also established (3) an indirect (via the images, *Label2Image2Grid*) mapping of the emotion terms to the valence–arousal space.

The valence and arousal ratings for the food images show good face validity: the highest mean valence ratings were obtained for the images of fresh fruit and pastries, while neutral ratings were obtained for the images of neutral foods and the lowest mean valence ratings correspond to the images of molded food.

The linkages between the terms and images have straightforward and meaningful interpretations: the food images that were overall rated low, near-neutral or high on valence were most frequently labeled with negative, neutral or positive emotion terms, respectively.

Although the relationship between the images and emotions terms was quite reliable (in the sense that it was a two-way mapping), the emotion terms were used more frequently to label the images in the *Label2Image* task than in the *Image2Label* task. This may be because the participants were more inclined to apply a given term to images in the *Label2Image* task, whereas they were less inclined to select the same term in the *Image2Label* task when they felt that it was less appropriate to characterize the image under consideration.

Note that the differences between the outcomes of the labeling and matching tasks may partly result from differences in terms of cognitive flow. Matching tasks that limit analytical thinking may tap more directly into the unconscious processes that are involved in food choice than the more cognitively demanding labeling tasks. The labeling tasks (*Label2Image* and *Label2Grid*) that were used in this study may be cognitively demanding since they required the participants to first imagine how a given (abstract) emotion term feels and then to either (a) select the foods (images) that seem capable to elicit that feeling (*Label2Image*) or (b) report the combination of valence and arousal that they associate with that term (Label2Image). The matching tasks (*Image2Label* and *Image2Grid*) may be cognitively less demanding than the labeling tasks since they required the participants to first imagine how consuming the food that is shown makes them feel (an intuitive response) and then to either (a) select the most appropriate labels to describe that feeling or (b) indicate their affective response on the EmojiGrid (intuitive graphical self-report tool). However, since cognitive flow is not a central topic of this study, we did not further investigate this issue here.

The valence and arousal values that were assigned to the emotion terms through indirect mapping to the valence–arousal space were typically less extreme than those that were assigned through direct mapping. Thus, the participants who imagined how a given emotion term felt in response to a given food (*Label2Grid* task) rated their feelings more extreme (intense) in terms of valence and arousal than the participants who imagined how the food that was shown in an image would make them feel (*Image2Grid* task). This may reflect the subjective nature of food perception: while all of the participants had a uniform notion of the emotion terms (e.g., ‘happy’), they differed in their appreciation of the food items that were represented in the images (e.g., strawberries can make someone ‘happy’, but most likely not everybody), resulting in a regression to the mean.

The indirect mapping that was obtained in this study shows a good overall agreement with the mapping that was reported by Jaeger et al. [34]. An interesting difference between both of the studies is the term guilty, which was mapped to the ‘activated displeasure’ domain in the study by Jaeger et al. [34] and to the ‘activated pleasure’ domain in this study. Figure 10 shows that the term guilty was most frequently associated with pastries, cookies, and the burger, items that are also most frequently associated with high-positive valence terms like happy, pleasant, and good. This result agrees with the observation that there is typically a cognitive association between guilt and hedonic pleasure [57]. These contrasting feelings, sometimes characterized as “guilty pleasures”, often coexist when we give in to a certain behavior (e.g., eating an appealing yet unhealthy food) that is known to have positive short-term but negative long-term consequences [58].

Single response tools that are based on a dimensional model of human core affect may be less appropriate when seeking detailed profiles of product-elicited emotional associations [34]. It has therefore been suggested to combine such tools with a CATA [59] or RATA [60] procedure. The results of this study suggest an implementation of the EmojiGrid that is similar to the Two-Dimensional Affect and Feeling Space (2DAFS [6]), where a core affect rating phase, in which items are rated on valence and arousal by clicking on a grid, is followed by an emotion-categorization phase in which only those emotion terms that are linked most strongly to the indicated position in the valence–arousal space are explicitly presented and selected (in a CATA paradigm) or rated (in a RATA paradigm) in order to increase sample discrimination. Note that an implementation of this kind could also be used to relate (translate) the responses from users from different cultures or language groups.

### 4.1. Limitations

The current study also has some limitations.

The CATA procedures that were used in the *Label2Image* and *Image2Label* tasks did not yield any information about the strength of the relations that are measured. Replacing the CATA by the RATA (rate-all-that apply) procedures could provide more insight into the degree to which the terms and images are related.

The EsSense Profile is purposefully dominated by emotion words with positive meanings in order to reflect the generally positive responses to commercial foods and beverages [12]. As a result, it is only sparsely populated with terms with more negative meanings. To achieve a denser coverage of the valence–arousal space with emotion words (especially on the low valence and low arousal sides), future studies could use lexicons that provide a more balanced list of positive and negative emotions [16,19].

Since the mapping between the images and the emotion lexicon (EsSense25) terms was derived via the mapping between the images and the valence–arousal space, it is not possible to distinguish between affective states with similar experimental features (e.g., similar valence and arousal).

In this study the mapping between the emotion terms and the valence–arousal space was only derived for UK participants and for a limited set of food images and emotion terms. Future studies should investigate a larger diversity in food images and more appropriate emotions terms. Other cultures may yield different mappings between images and words. Hence, the results may not extrapolate to other groups, different images or different terms.

### 4.2. Future Research

Future studies may also investigate the mapping between emotion terms and the valence–arousal space through experiments in which food or beverages are actually tasted or consumed instead of merely visually perceived. Affective appraisal of food images is a ‘cold’ cognitive evaluation process that is based on criteria reflecting personal experiences and relevance [61]. Previous research has shown that viewing food pictures activates brain areas that code how the food that is perceived tastes (the insula/operculum) and how rewarding it would be to eat it (the orbitofrontal cortex; [62,63]). Hence, people can reliably produce an affect rating without actually tasting the food that is shown. However, while the results from a tasting experiment will most likely agree with our present results for the valence dimension (which are typically quite stable and consensual across experimental paradigms [61]), they may differ for the arousal dimension as a result of the sensory characteristics of the sample and a higher degree of personal relevance when it is tasted or consumed [64].

## Figures and Tables

**Figure 1 foods-11-00972-f001:**
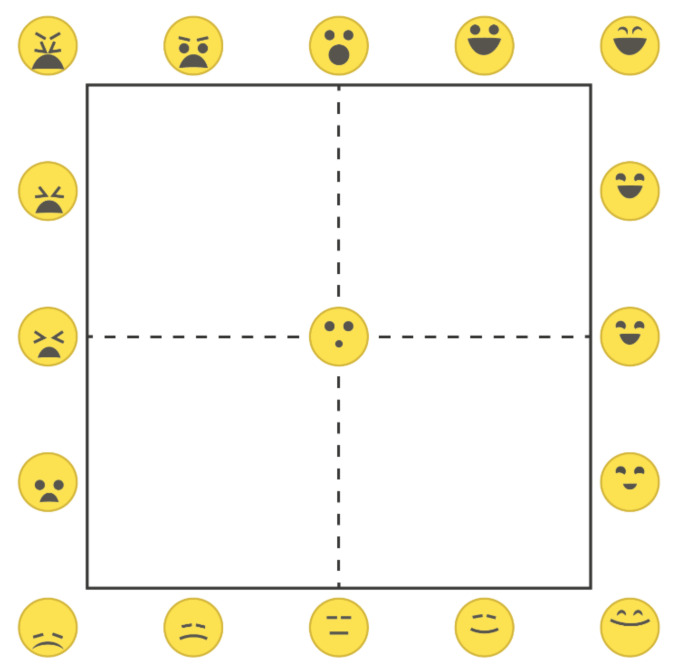
The EmojiGrid (from [20], see also https://en.wikipedia.org/wiki/EmojiGrid, (accessed on 4 March 2022). The x-axis represents the valence rating, whereas the y-axis represents the arousal rating, both on a scale from 0–100.

**Figure 2 foods-11-00972-f002:**
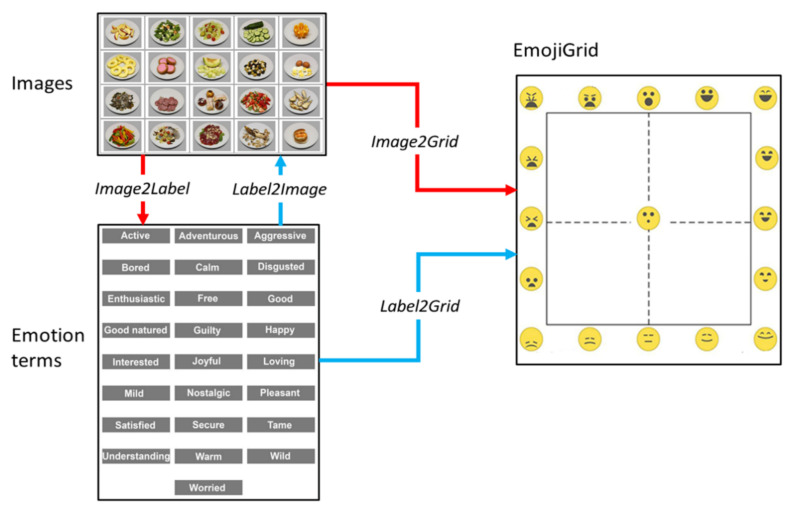
The four different tasks (represented by the arrows) that were investigated in this study. Red arrows: matching tasks. Blue arrows: labeling tasks. *Image2Grid* task: matching food images to the EmojiGrid. *Image2Label* task: matching food images to emotion terms. *Label2Image* task: labeling food images with emotion terms. *Label2Grid* task: labeling the EmojiGrid with emotion terms.

**Figure 3 foods-11-00972-f003:**
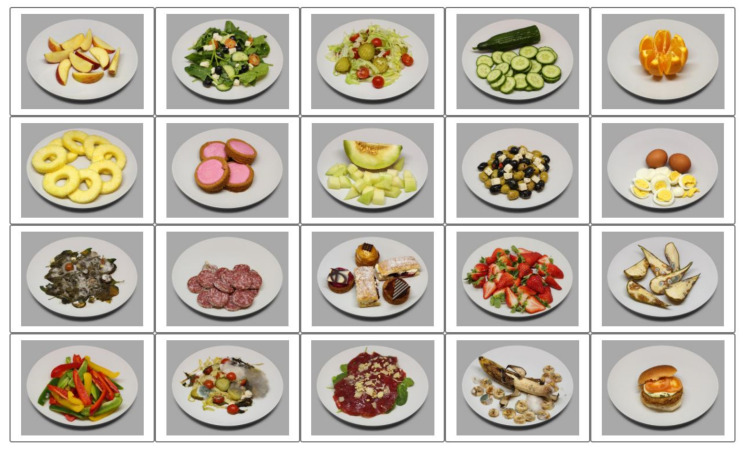
The set of 20 food images used as stimuli in this study (selected from the CROCUFID database [40]).

**Figure 4 foods-11-00972-f004:**
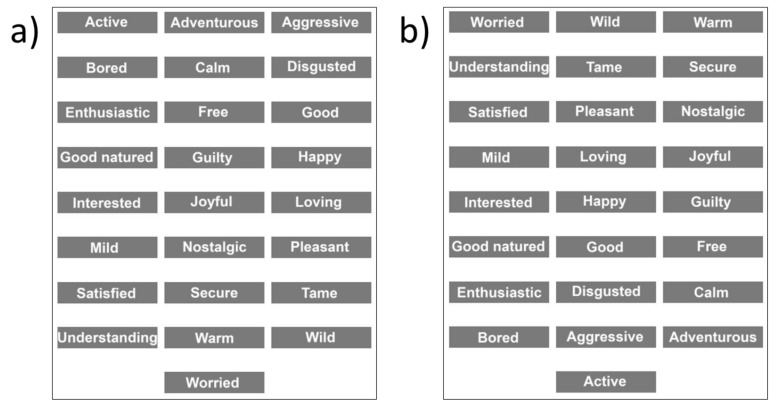
Screenshots of the EsSense25 word list in (**a**) alphabetical and (**b**) reversed order (from [41]).

**Figure 5 foods-11-00972-f005:**
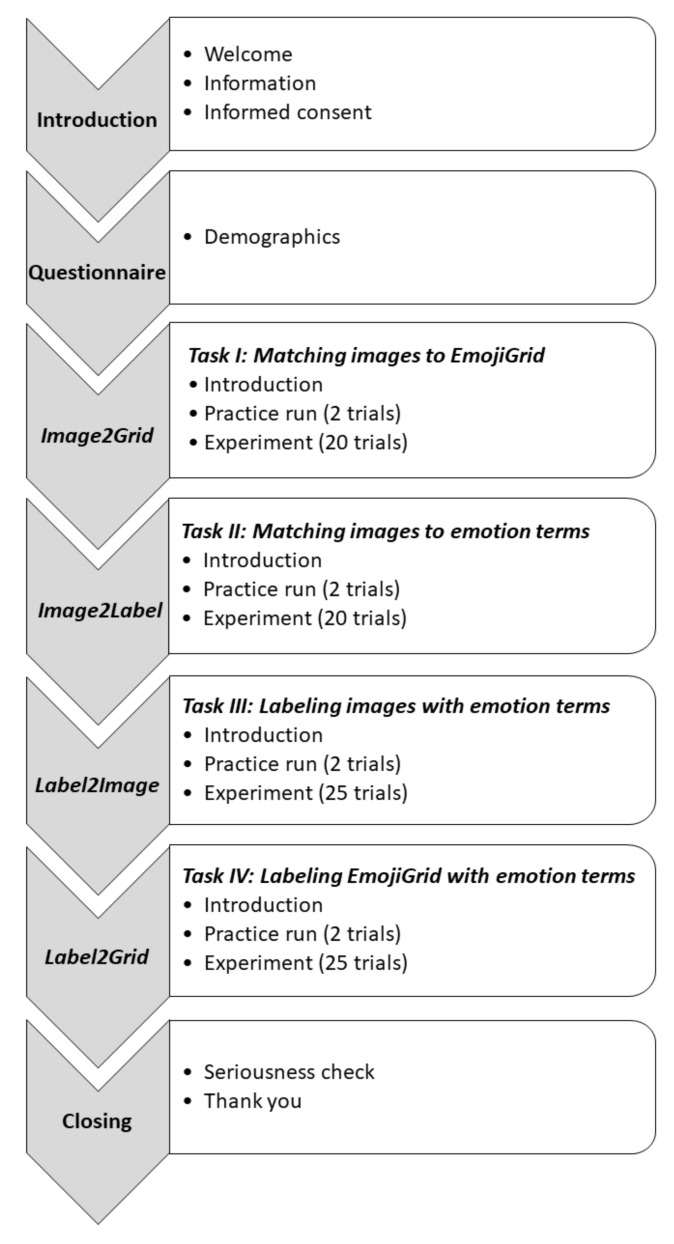
Schematic representation of the experimental procedure.

**Figure 6 foods-11-00972-f006:**
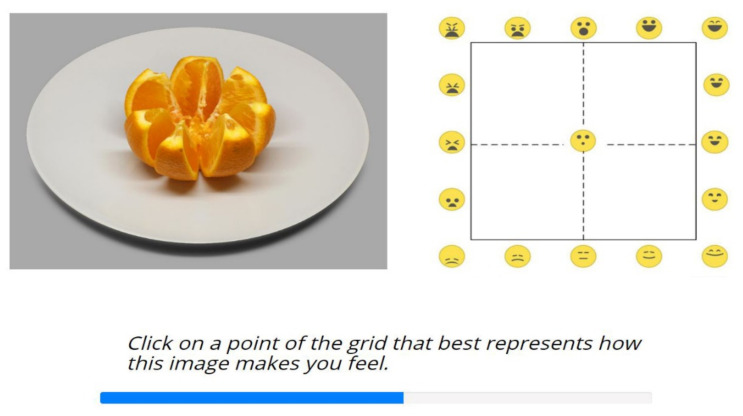
Screen layout of the *Image2Grid* task: mapping food images to the EmojiGrid. A blue progress bar indicated the progression of the task.

**Figure 7 foods-11-00972-f007:**
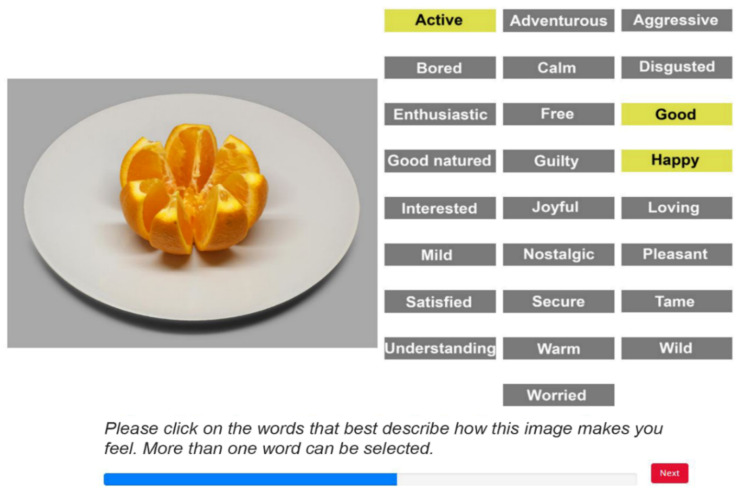
Screen layout of the *Image2Label* task: mapping food images to EsSense25 terms. Selected terms became highlighted in yellow. In this example the participant responded “active”, “good” and “happy”. A blue progress bar indicated the progression of the task.

**Figure 8 foods-11-00972-f008:**
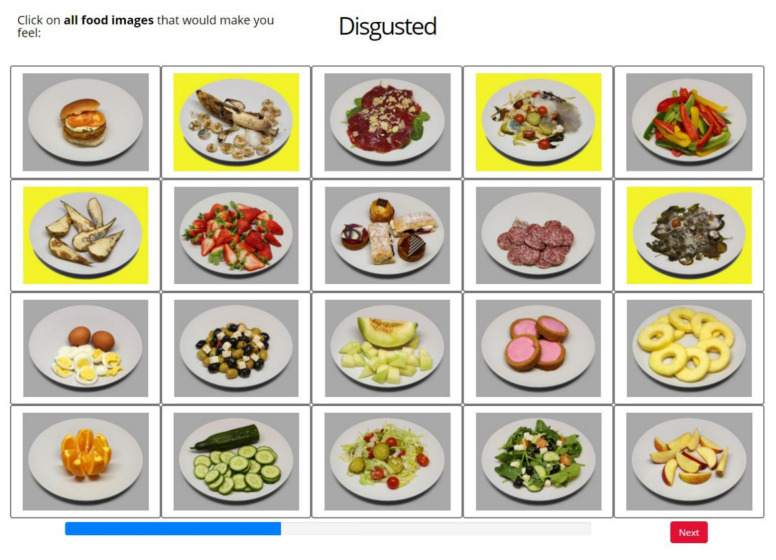
Screen layout of the *Label2Image* task: labeling food images with emotion terms (in this example the term “disgusted”). Images with a yellow background were selected by the participant. A blue progress bar indicated the progression of the task.

**Figure 9 foods-11-00972-f009:**
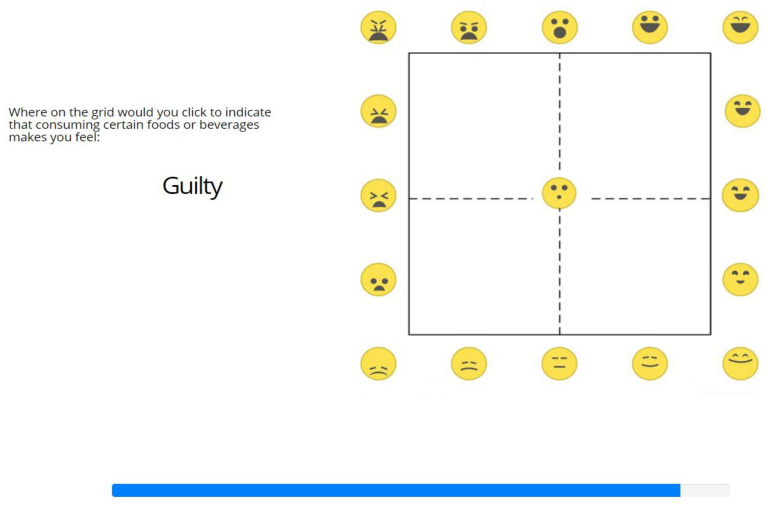
Screen layout of the *Label2Grid* task: mapping EsSense25 terms (in this example, the term “guilty”) to the EmojiGrid.

**Figure 10 foods-11-00972-f010:**
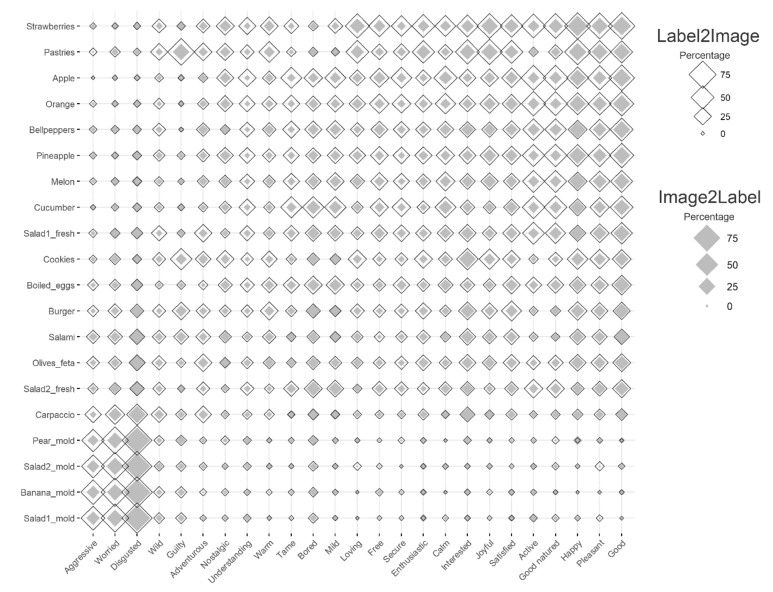
Percentage (represented by symbol size, ranging between 0% and 100%) of participants (*n* = 480) that linked each image to a subset of the EsSense25 terms (Image2Label task, filled diamonds), and each EsSense25 term to a subset of the food images (*Label2Image* task, open diamonds). Emotions terms and food images are arranged in increasing average valence order along the horizontal and vertical axes.

**Figure 11 foods-11-00972-f011:**
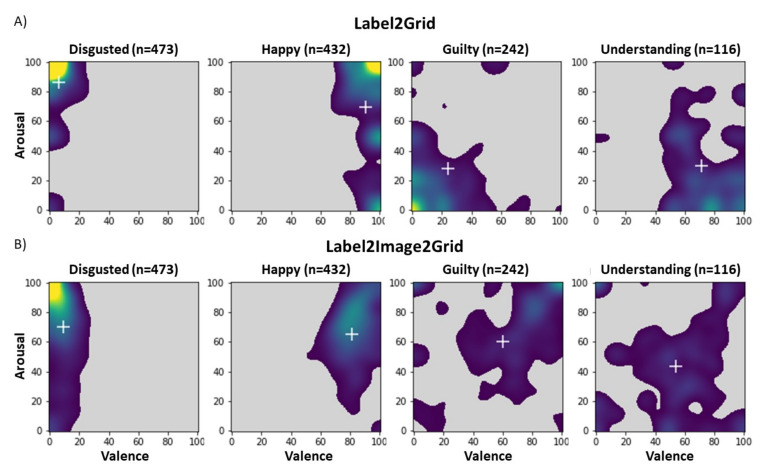
Distribution of the mapping responses over the valence–arousal space, for four different words (disgusted, happy, guilty, and understanding). (**A**) Distribution of the direct mapping responses, when participants were asked to map a word directly to the EmojiGrid (*Label2Grid*). (**B**) Distribution of the indirect (*Label2Image2Grid*) term-to-EmojiGrid mapping responses, when participants first linked a term to an image (*Label2Image* task) and then rated the image in terms of valence and arousal (*Image2Grid* task). Crosses signify group mean arousal and valence ratings.

**Figure 12 foods-11-00972-f012:**
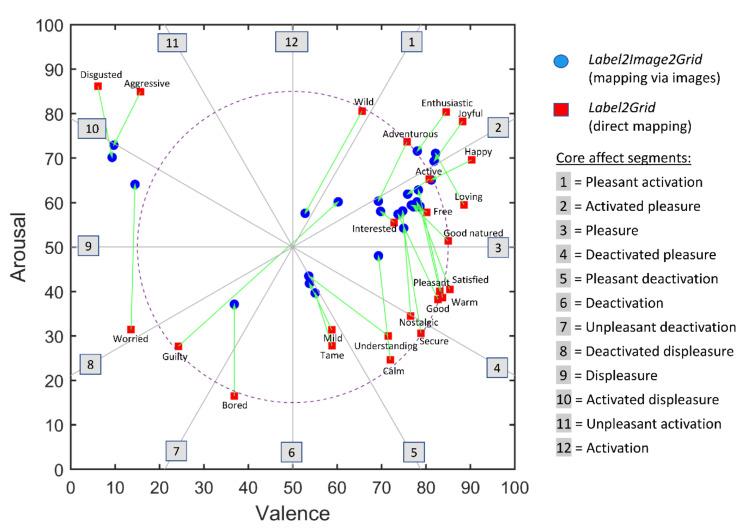
Mapping the EsSense25 terms to the EmojiGrid. Red symbols represent the average positions of the emotion terms that were directly mapped to the EmojiGrid in *Label2Grid* task. Blue symbols represent the average positions of the emotion terms that were mapped to the EmojiGrid via the food images (*Label2Image* task followed by *Image2Grid* task). Green arrows connect corresponding terms obtained through direct and indirect mapping. The gray labels correspond to the 12 domains (delineated by the gray radial axes) of the core affect model presented by Yik et al. [27]. The dashed circle serves as a visual reference for the eccentricity of the data points.

**Table 1 foods-11-00972-t001:** The mean valence (V) and arousal (A) ratings resulting from the *Image2Grid* task and the corresponding values provided in the CROCUFID database (Vc, Ac; [40]) for each of the 20 selected food images (ID is the original image identifier in the CROCUFID database, the index c refers to values from the CROCUFID database).

ID	Food Image	V	A	Vc	Ac
123	Salad1_mold	6.71	78.51	5.11	86.36
190	Banana_mold	7.09	76.80	6.77	75.80
167	Salad2_mold	8.24	76.65	6.62	83.15
152	Pear_mold	9.05	72.41	8.43	74.89
175	Carpaccio	38.09	46.51	34.92	53.79
13	Salad2_fresh	48.61	46.25	53.13	44.10
82	Olives_feta	48.65	56.92	47.61	58.79
136	Salami	51.65	51.07	42.23	52.03
250	Burger	57.96	47.95	58.08	49.44
93	Boiled_eggs	58.06	47.49	58.31	46.31
47	Cookies	58.94	50.17	63.05	50.39
9	Salad1_fresh	62.05	51.11	60.97	48.90
36	Cucumber	62.82	46.36	62.67	50.57
70	Melon	64.20	54.04	63.74	54.00
44	Pineapple	66.69	53.80	70.39	60.70
162	Bellpeppers	67.88	50.58	70.62	50.56
43	Orange	70.57	51.73	70.77	55.10
4	Apple	74.18	49.43	66.92	54.16
145	Pastries	77.62	65.69	79.07	65.89
147	Strawberries	79.62	67.50	80.85	64.95

**Table 2 foods-11-00972-t002:** Mean valence and arousal ratings for each emotion term, determined from the direct (*Label2Grid* or L2G) and indirect (*Label2Image2Grid* or L2I2G) mapping of the terms to the valence–arousal space, together with the results of the Wilcoxon signed-rank test. Bold indicates a significant difference between the indirect and direct mappings. Here, *n* represents the number of participants that used the emotion terms at least once.

		Valence			Arousal		
	*n*	L2G	L2I2G	*p*	L2G	L2I2G	*p*
Understanding	116	71.48	53.60	**<0.001**	30.03	43.47	**<0.001**
Wild	132	65.62	52.77	**<0.001**	80.58	57.58	**<0.001**
Secure	181	78.89	74.97	**0.147**	30.62	54.24	**<0.001**
Aggressive	186	15.68	9.68	**<0.001**	84.88	72.90	**<0.001**
Tame	190	58.83	55.03	0.0510	27.76	39.67	**<0.001**
Adventurous	206	75.73	69.28	**<0.001**	73.64	60.32	**<0.001**
Active	213	80.76	75.86	**<0.005**	65.31	61.89	0.211
Warm	230	83.74	78.29	**<0.001**	38.60	62.83	**<0.001**
Free	238	80.21	76.70	**<0.001**	57.86	59.44	0.656
Guilty	242	24.24	60.23	**<0.001**	27.69	60.17	**<0.001**
Loving	247	88.57	82.15	**<0.001**	59.48	71.06	**<0.001**
Enthusiastic	258	84.53	78.06	**<0.001**	80.38	71.58	**<0.001**
Nostalgic	261	76.56	74.75	0.199	34.51	58.11	**<0.001**
Good	277	82.72	73.70	**<0.001**	38.21	57.39	**<0.001**
Calm	306	71.96	69.30	0.108	24.59	48.02	**<0.001**
Mild	330	58.79	53.75	**<0.005**	31.36	41.82	**<0.001**
Satisfied	334	85.37	77.96	**<0.001**	40.49	60.13	**<0.001**
Worried	340	13.56	14.46	0.967	31.47	64.09	**<0.001**
Joyful	363	88.26	81.83	**<0.001**	78.26	69.31	**<0.001**
Bored	364	36.81	36.83	0.635	16.51	37.13	**<0.001**
Interested	377	72.80	69.77	**0.379**	55.57	58.03	0.213
Pleasant	408	83.14	78.66	**<0.001**	40.04	59.14	**<0.001**
Good natured	428	84.98	77.24	**<0.001**	51.40	59.01	**<0.001**
Happy	432	90.27	81.21	**<0.001**	69.58	65.09	**<0.001**
Disgusted	473	6.11	9.31	**<0.001**	86.17	7.16	**<0.001**

**Table 3 foods-11-00972-t003:** Mapping between the EsSense25 emotion terms and the 12 domains of the core affect model presented by Yik et al. [27], from the study by Jaeger et al. [34] and obtained in this study through both direct mapping (*Label2Grid* task) and indirect mapping (*Label2Image2Grid*). Fractional numbers correspond to relative positions within domains (see text). Numbers in boldface indicate mappings obtained in this study that consistently differed more than two domains from those reported by Jaeger et al. [34].

	Core Affect Domain		
Emotion Term	Jaeger et al. [34]	Indirect Mapping	Direct Mapping
Adventurous	1	3.1	2.6
Active	1	**3.2**	**3.1**
Wild	1.5	1.7	1.9
Enthusiastic	1.5	2.7	2.6
Free	1.5	3.4	3.5
Loving	2	2.9	3.5
Joyful	2	3.0	2.8
Happy	3	3.1	3.1
Interested	3	3.3	3.5
Good natured	3	3.4	3.9
Pleasant	3	3.4	4.6
Good	3	3.4	4.7
Satisfied	4	3.3	4.5
Secure	4	3.7	5.1
Warm	4.5	3.2	4.6
Nostalgic	4.5	3.4	5.0
Understanding	4.5	6.0	5.4
Mild	4.5	6.2	6.2
Calm	5	4.2	5.6
Tame	6	6.1	6.3
Bored	7	8.5	9.3
Guilty	10	**2.5**	**2.6**
Disgusted	10	10.9	11.3
Aggressive	10.5	11.0	11.5
Worried	11	10.7	9.1

## Data Availability

Data is contained within the article and Appendix A.

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
