# Peer review of "Linking Categorical and Dimensional Approaches to Assess Food-Related Emotions"

_foods, 2022, doi:10.3390/foods11070972_

Round 1
Reviewer 1 Report
There are only some minor mistakes which have to be changed:
There are two Figures 1, after Figure 5 comes Figure 2, after Figure 7 there is Figure 3 - seems a bit chaotic - please change that.
There are several error references in the text (88, 232, 279, 313, 346, 377)
Please describe whether participants have been asked to actively imagine to eat the food shown on the pictures or not? What was the exact instruction?
The authors are correct in their assessment, that the EsSense25 system includes too little negative emotions. This became immediately clear to me when I saw the orange fruit in Fig. 7 - I would be annoyed by the strange way the fruit has been cut - probably resulting in a mess when eating it.
Author Response
First of all, we would like to thank you for your feedback on our work.
Comment : There are two Figures 1, after Figure 5 comes Figure 2, after Figure 7 there is Figure 3 - seems a bit chaotic - please change that.
Authors’ reply: Unfortunately the software used by MDPI to convert the Word document we submitted into the MDPI format, introduced many errors, such as incorrect numbering of figures, additional and duplicate lines in the text, conversion of super- in subscripts, etc. It seems that the MDPI conversion process cannot handle automatic numbering (field codes). In the Word document which we submitted the figures were numbered correctly, and all in-text references were correct. We (hopefully) resolved this issue by removing all field codes from our Word document before we submitted the revised version.
Comment :There are several error references in the text (88, 232, 279, 313, 346, 377)
Authors’ reply: see our reply above: these errors were introduced by the journal’s automatic file conversion process, and are (hopefully) resolved now by removing all field codes from our document
Comment :Please describe whether participants have been asked to actively imagine to eat the food shown on the pictures or not? What was the exact instruction?
Authors’ reply: In the introductory information for this experiment, we informed the participants that they would be asked to rate how consuming the food shown in various pictures would make them feel. We now added this information in the procedure section.
Reviewer 2 Report
The manuscript explores how three different tasks captures emotional reactions.
There are a lot of dataset collected in this study, yet the flow in the results isn't so intuitive. It was clear that the authors attempted to see how the tasks goes for 1-4, yet only label2grid and label2image is compared?
The trial is rather long, how did the authors ensure that the participants arent fatigued?
What would be interesting for Fig 4 is to see how these 2 methods differ. Is it possible to add-on some form of sig. test on each diamond nodes that are significant?
Fig 5. What does the + imply? mean?
Fig 6, to measure the distance of the two methods - wouldn't superimposed MFA plot work better to further dissect the distance on how each emotional terms differ? Or is if the authors wants to keep the plot to include the core affect, is it possible for the authors to then measure the cosine angles of each points and compare them? Some form of MDA can be applied to further decipher which emotional terms are better or poorly described?
The discussion should also include an explanation on how different tasks differ in terms of cognitive flow.
Author Response
First of all, we would like to thank you for your feedback on our work.
Comment :There are a lot of dataset collected in this study, yet the flow in the results isn't so intuitive. It was clear that the authors attempted to see how the tasks goes for 1-4, yet only label2grid and label2image is compared?
Authors’ reply: Image2Label and Label2Image are directly compared. Also, the direct mapping of labels to the grid (Label2Grid) is compared to the indirect mapping of labels to the grid via the images (Label2Image2Grid = Label2Image followed by Image2Grid). Hence all four tasks are relevant for - and used in - these comparisons.
Comment :The trial is rather long, how did the authors ensure that the participants arent fatigued?
Authors’ reply: The average duration of the experiment was only about 15 minutes, which is actually quite short for this type of experiment. We now explicitly mention the duration of the experiment in the method section.
Comment :What would be interesting for Fig 4 is to see how these 2 methods differ. Is it possible to add-on some form of sig. test on each diamond nodes that are significant?
Authors’ reply: We agree that it would be interesting to see where the two methods lead to significantly different results. However, due to the high number of comparisons (500), the alpha level for these tests should be adjusted and then become too low for a (non-parametric) test to lead to useful results.
Comment :Fig 5. What does the + imply? mean?
Authors’ reply: Unfortunately the submission system messed up all figure numberings in our word file (see our reply to reviewer 1). The reviewer is pointing to Figure 5, but it seems likely that Figure 11 is meant here. In Figure 11, the + signs in the distribution plots represent the group mean arousal and valence ratings (as explained in the caption of this figure).
Comment :Fig 6, to measure the distance of the two methods - wouldn't superimposed MFA plot work better to further dissect the distance on how each emotional terms differ? Or is if the authors wants to keep the plot to include the core affect, is it possible for the authors to then measure the cosine angles of each points and compare them? Some form of MDA can be applied to further decipher which emotional terms are better or poorly described?
Authors’ reply:
The use of MFA is in this case superfluous because the emotional terms are described using only two dimensions – arousal and valence. As such, any embedding obtained by e.g. MFA will result in a reconstructed substitute of the space that we in fact have already obtained through direct mapping of the emotional terms to the valence-arousal space. Therefore, using this direct mapping of the emotional terms for both label2Grid and label2Image2Grid terms preserves the original aspects of affect better than would a reconstructed space (e.g. the relative orientation of points is not dependent on the embedding but directly related to affective dimensions).
Comment :The discussion should also include an explanation on how different tasks differ in terms of cognitive flow.
Authors’ reply:
While cognitive flow the topic of this study and not our main interest here, we now mention in the Limitations section that these tasks may differ in terms of cognitive flow:
“ Note that differences between the outcome of the labeling and matching tasks may partly result from differences in terms of cognitive flow. Matching tasks that limit analytical thinking may tap more directly into unconscious processes involved in food choice than more cognitively demanding labeling tasks. The labeling tasks (Label2Image, Label2Grid) used in this study may be cognitively demanding since they require participants to first imagine how it is to feel like a given (abstract) emotion term and then either (a) to select the foods (images) that seem capable to elicit that feeling (Label2Image) or (b) to report the combination of valence and arousal they associate with that term (Label2Image). The matching tasks (Image2Label, Image2Grid) may be cognitively less demanding than the labeling tasks since they require participants to first imagine how consuming the food that is shown makes them feel (an intuitive response) and then either to select the most appropriate labels to describe that feeling or to indicate their affective response on EmojiGrid (intuitive graphical self-report tool). However, since cognitive flow is not a central topic of this study we did not further investigate this issue here.”
Round 2
Reviewer 2 Report
The authors have addressed the comments